# Technology of Orodispersible Polymer Films with Micronized Loratadine—Influence of Different Drug Loadings on Film Properties

**DOI:** 10.3390/pharmaceutics12030250

**Published:** 2020-03-10

**Authors:** Katarzyna Centkowska, Elżbieta Ławrecka, Malgorzata Sznitowska

**Affiliations:** Department of Pharmaceutical Technology, Medical University of Gdansk, Hallera Str. 107, 80-416 Gdansk, Poland

**Keywords:** orodispersible film, suspension, loratadine, mechanical properties, viscosity, content uniformity

## Abstract

The production of orodispersible films (ODFs) with suspended insoluble drug substances is still a challenge, mainly due to the difficulty associated with achieving a proper homogeneity and mechanical properties of the films. Hypromellose (HPMC) and a mixture of polyvinyl alcohol (AP) and povidone (PVP) were compared in terms of their suitability for ODFs incorporating suspended micronized loratadine (LO) in a concentration range of 10%–40%. In a planetary mixer (Thinky), a uniform dispersion of LO in an aqueous viscous casting solution was obtained. The suspended LO particles caused dose-dependent changes in the viscosity of the casting mass and affected the mechanical quality of ODFs. Drug concentrations higher than 30% reduced the film flexibility and tear resistance, depending on the polymer type. LO films with a thickness of 100 µm disintegrated within 60-100 s, with no significant influence of the LO content in the range 10%–30%. HPMC films, regardless of the drug concentration, met the pharmacopoeial requirements regarding the uniformity of the drug content. AP/PVP films were too elastic, and the drug content uniformity was not achieved. The conclusion is that, using an HPMC matrix, it is possible to obtain a high load of a poorly water-soluble drug (30% of dry film mass corresponds to a dose of 5 mg per 1.5 cm^2^) in ODFs characterized by proper physical characteristics.

## 1. Introduction

Orodispersible polymer films (ODFs) are a relatively novel dosage form, designed to be applied to the oral cavity. They are described in the European Pharmacopoeia (Ph. Eur. 10th) monograph “Oromucosal preparations” [1]. Due to the fact that the active substance can be administered in a solid form, but without the need to swallow them with water, they are a good alternative as oral drugs for special patient groups, e.g., geriatric or pediatric patients and patients with dysphagia [2,3,4,5,6]. The film is hydrated by saliva when placed on the tongue and disintegrates rapidly, releasing the active substance for local (oromucosal) or systemic (gastrointestinal) absorption, without the risk of choking [7]. This does not mean, however, that the drug itself dissolves quickly, and for poorly soluble drugs, it can be still a challenge to obtain fast dissolution and absorption [8,9,10].

ODFs are manufactured using water-soluble polymers as matrix components [11]. The most common examples are hydrophilic polymers: Hypromellose (HPMC), hydroxypropyl cellulose (HPC), maltodextrin (MDX), and pullulan (PU), with a molecular weight between 1000 and 9000 Da [11,12]. For the film preparation, the solvent casting method is a typical technique, which has been described well [13,14,15]. The main problems connected with the manufacturing process and batch-to-batch variability include entrapped air bubbles and an insufficient uniformity of the mass or drug content, which often results from an improper viscosity, and problems associated with a good mass spreadability during casting [15,16]. To ensure the accurate drug dose in a single ODF is an easier task if the drug is dissolved in the casted polymer solution. Thus, ODF technology is currently mainly used for water-soluble active substances, and the most desired situation is that, after solvent evaporation, the drug does not crystallize and remains dissolved in the ODF matrix [15], although among the marketed ODFs, ondansetron and sildenafil citrate films contain these hydrophilic drugs in a dispersed phase, which is formed after the evaporation of water. If the active drug substance (API—active pharmaceutical ingredient) is insoluble in water, a common procedure is to use organic solvents in the production process [9,10,15,17,18,19,20]. However, in such a situation, drug recrystallization is even more probable, which results in a problematic bioavailability.

The other possibility is to suspend the drug in the aqueous solution of the polymer used for casting the film, which allows uncontrolled recrystallization to be avoided [8,16,17,19,20,21,22,23,24]. However, the formulation of such biphasic films, containing suspended API, implies serious challenges regarding the uniformity of the content, texture, and appropriate mechanical properties of the films [15]. For example, ODFs with ibuprofen, incorporated as undissolved particles (8.3 mg/cm^2^), demonstrated a rough surface and high thickness [17].

In our study, a poorly water-soluble drug, loratadine (BCS class II), was incorporated as micronized crystals into two different orodispersible film matrices. Loratadine (LO) is a commonly prescribed antihistamine drug for the treatment of allergy events not only in adults, but also in pediatric patients above two years old [25,26].

In Europe, for children below six years of age, loratadine is available only in liquid forms: syrup or suspension (5 mg/5 mL), while in the USA, chewable tablets have recently been introduced (Claritin chewables 5 mg, Bayer). Orodipersible tablets (RediTabs) containing 5 mg or 10 mg of loratadine are registered only for children above six years. With a growing interest in orodispersible formulations, ODF can be a good alternative, especially if it allows for dose adjustment by dividing the film into appropriate parts [27].

The solubility of loratadine (LO) in water is reported as 10 µg/mL or lower [28,29], and the taste is not so problematic as for water-soluble drugs. The successful development of ODF with LO in suspension may also be helpful for the design of similar formulations with other poorly soluble APIs.

The aim of the research was to investigate the influence of the composition and viscosity of the polymer solution, as well as the dispersion technique, on the mechanical properties, homogeneity, and disintegration of the obtained ODFs, containing different amounts of LO. One of the problems that should be solved for the suspension-type ODF is the surface roughness. Thus, we propose to use micronized LO, with a particle size below 10 µm, to achieve a smooth surface film structure, with a good palatability. 

The LO dose for children with a body weight less than 30 kg is 5 mg once daily [30]. Because, in our opinion, an acceptable area of ODF proposed for small children should not be larger than 3 cm^2^, the aim was to incorporate at least 5 mg of LO in ODF of this size.

## 2. Materials and Methods 

### 2.1. Materials

Loratadine (LO, Cadila Pharmaceuticals Ltd., Ahmedabad, India) was a gift from PPF Hasco-Lek (Wroclaw, Poland). The polymers used to prepare ODF matrix were: Hypromellose (HPMC, Pharmacoat 606, Shin-Etsu Chemical, Tokyo, Japan), two types (AP40 and AP05) of polyvinyl alcohol (Gohsenol EG-40PW and EG-05PW from Nippon Synthetic Chemical Industries, Tokyo, Japan), and polyvinylpyrrolidone (PVP, Kollidon 30, BASF, Ludwigshafen, Germany). Polyethylene glycol (PEG, Macrogol 400, Aldrich Chemistry, Saint Louis, MO, USA) was employed as a plasticizer. Purified pharmaceutical-grade water was used as a solvent. Triethylamine buffer (0.1%), adjusted to pH 4.0 with acetic acid, was used for HPLC analysis. Acetonitryl (Merck, Darmstadt, Germany) and methanol (POCh, Gliwice, Poland) were of HPLC grade. All other reagents were of analytical grade.

### 2.2. Preparation of the Casting Suspension and ODFs

Polymers (15.0% *w/w* hypromellose or a mixture of 7.5% *w/w* polyvinyl alcohol and 7.5% *w/w* polyvinylpyrrolidone) were dissolved in water with a plasticizer (3.0% *w/w* PEG) under constant stirring (Heidolph type MR 30001K, Schwebach, Germany) at room temperature (HPMC) or at 60°C (AP/PVP). The LO (2.0%–12.0% *w/w*) was added to achieve a final content of 10% to 40% (*w/w*) in dry ODFs. The suspensions were homogenized using three different techniques: 1) A planetary centrifugal mixer (Thinky ARE-250, Thinky USA Inc., Laguna Hills, CA, USA) in defoaming mode (15 min at 2200 rpm); 2) a higher shear mixer, Ultra Turrax T 25 D (IKA, Staufen, Germany), equipped with a dispersing tool, S25N10G (5 min at 12 000 rpm); and 3) a direct probe-type ultrasonic disintegrator (Techpan, Ultrasonic disintegrator type UD-20, Warsaw, Poland) (10 min at 300 W). The suspensions were de-aerated in a vacuum chamber system (Thermo Fischer Scientific, Waltham, MA, USA). 

Drug-free and drug-loaded films were prepared by a solvent casting method. A film applicator (Camag, Muttenz, Switzerland) was used, and the liquid mass was casted onto glass plates (20 cm × 20 cm), with different casting heights: 300 µm or 800 µm. The casted films were dried for 24 h at room temperature, followed by 1 h at 37 °C. Pieces of 3 cm^2^ (1.5 cm × 2 cm) were cut and stored in a monitored room environment (20–25 °C and 40–50% RH).

### 2.3. Homogeneity of the Casting Suspension and ODFs

The mass of the film (*n* = 10) per 3 cm^2^ area was measured (analytical balance, WAXX 6, Radwag, Radom, Poland). The thickness of the ODFs (*n* = 10) was measured with a thickness gauge (ElektroPhysik MiniTest 730, Cologne, Germany) at three points for each 3 cm^2^ fragment. 

The macroscopic characteristics of the ODFs consisted of the following observations: Roughness of the surface, color homogeneity, flexibility, and ability to be removed from the glass surface film after drying. The casting suspensions and film surface were also examined by a light stereoscopic microscope (Opta-Tech X 2000, Opta-Tech, Warsaw, Poland), equipped with a digital camera (Panasonic type GP - KR 222E from Matsuhita Electric Industrial, Osaka, Japan). The size of the particles in suspensions was measured using Opta View 1 (version 7.1.0.4) software (Opta-Tech, Warsaw, Poland).

### 2.4. Morphological Properties

The ODF surface analysis was performed using atomic force microscopy (Nanosurf Flex-Axiom, Liestal, Switzerland), equipped with motorized translator ATS 204 (Nanosurf AG, Liestal, Switzerland) and the Nanosurf C3000 controller program. A scan head (PPP-EFM, Nanosensors, Neuchatel, Switzerland) was used for imaging, with an operating-mode phase contrast error range of 20V. Images of 60 × 60 µm in size were made in an air environment. Nanosurf (version 3.8.0.8) and Gwiddion (version 2.49) softwares (Nanosurf Flex-Axiom, Liestal, Switzerland) were used for image processing.

The ODF surface analysis was also performed using a digital microscope (Keyence VHX 6000, Keyance International NV/SA, Mechelen, Belgium), equipped with a high-performance zoom lens, Z20T (20x-200x). 3D visualization on the basis of high-resolution 2D images (HDR) was conducted.

### 2.5. Viscosity of the Casting Mass

The viscosity of the polymer matrix solutions and LO suspensions was determined (*n* = 3) with a rotational viscometer (ViscoTester VT550 Haake, Thermo Scientific, Karlsruhe, Germany), equipped with a cone-plate system (PK 1, cone radius 14 mm, cone angle 1°). The test was performed at a shear rate of 10 s^−1^ at 20 ± 1 °C.

### 2.6. Water Content

The water content was determined using a Karl–Fisher potentiometric titration apparatus (Mettler Toledo DL 38, Greifensee, Switzerland). About 150–200 mg of a film was cut into small pieces, weighed, and introduced, with a weighing boat, directly to the titration vessel. The samples were stirred before measurements for 180 s. A two-component reagent (Merck Aquara Combi Titrant 2, Darmstadt, Germany) was used.

### 2.7. Disintegration Time

The single-dose films were placed in a glass Petri dish (5 cm^2^) in 10 mL of purified water at 37 °C. The plate was swirled manually every 5 s, until the ODF dissolved or disintegrated into the suspension. The time was recorded, and the average of 3 samples was calculated.

### 2.8. Viscoelastic Properties

The mechanical properties were analyzed with a texture analyzer (TA.XT plus, Stable Micro Systems, Godalming, UK). Strips of the films of 10 mm wide were fixed in tensile grips, placed at an initial distance of 20 mm. The grips were extended in a tension mode, with a test speed of 0.5 mm/s, and set to the maximum force (break point), with a cell loading of 5 kg. The measurements were repeated for at least six samples. The tear resistance (TR) (N) was measured as the maximum force required to rupture or tear the film, and the tensile strength (maximum force, divided by the cross-section area of the film, TS) (N/mm^2^)and elongation at the break (increase in the length, divided by the initial film length, multiplied by 100, %E) (%) were calculated [31]. Young’s modulus (E) was determined from a slope of the linear region of the stress–strain curve. All experiments were conducted under ambient conditions (20–25 °C, 40–50% RH). For the evaluation, the arithmetic mean and standard deviation (SD) for at least 4 samples were calculated.

### 2.9. Folding Endurance

The folding endurance (FE) was examined by folding each film (*n* = 3) at the same place, until it broke. The number of folds required to break the ODFs or to develop a visible crack was determined [32].

### 2.10. Uniformity of Content

The uniformity of the LO content in the ODFs was carried out according to the Ph. Eur. [1] monograph, “Uniformity of dosage units” (test A for solid dosage forms). Ten ODFs were analyzed. They were cut from the larger film casted on a glass plate. The LO content in each film was determined using the HPLC method (HPLC Merck-Hitachi, Darmstadt, Germany), with a reversed phase C18 column (5 μm LiChrosphere 250 × 4 mm, Merck) and UV–Vis detector set to 254 nm. A mixture of 0.1% *w/w* of a triethylamine buffer, pH 4.0, and acetonitrile (60:40 *v/v*) was used as a mobile phase. The injection volume and flow rate of the eluent were 20 µL and 2 mL/min, respectively. A linear response was obtained in a range from 1 to 100 μg/mL of the LO solution. Each film (size 3 cm^2^) was weighed and dissolved in a methanol/water mixture (1:2.5 *v/v*). An appropriate volume of the solvent, depending on the polymer type, was used: 35.0 mL for HPMC and 50 mL for AP/PVP ODFs. Immediately after the dissolution, the samples were diluted, with a mobile phase of 1:10, and analyzed. 

The LO content in a single-dose ODF was expressed as the % of a theoretical content, and the acceptance value (AV) was calculated according to Ph. Eur. [1].

### 2.11. Statistical Analysis

The results were expressed as the mean value ± SD. The data, illustrating the influence of the LO content on the disintegration time and on the uniformity of the dosage, were analyzed by a t-test and one-way ANOVA test using the STATISTICA 13.1 software (TIBCO Software, Palo Alto, CA, USA). A statistical p-value of less than 0.05 was considered significant.

## 3. Results

### 3.1. Viscosity of the Casting Mass and Morphology of ODFs

Two types of polymer matrices were compared in the research: A commonly used HPMC (type 606) and a new composition consisting of an AP and PVP mixture. AP and PVP have recently been used alone as polymers forming an ODF matrix. However, more commonly, they serve as additives, modifying the properties of matrices from other polymers, e.g., HPC and HPMC [11,33,34,35]. A PVP with a low molecular weight (PVP K30, m.w. 29 000) was used in the formulation, although other researchers reported the use of a PVP with a higher molecular weight (90 000–360 000 Da) in ODF formulations [11,36,37,38]. 

The composition of the ODF matrix was selected based on the tests performed for the placebo ODF. The big problem when designing ODFs is the a priori determination of the range of the optimal viscosity of the casting mass. Moreover, their viscosity, although very important, is not the only factor determining the performance of the mass during the film formation, because other parameters, related to the structure of the polymer chains, interactions between the components, the type of solvent, or the surface tension also have an impact. Additional changes in the properties of the casting mass result from the presence of API. In the suspension-type system, formed by an insoluble API, there are additional problems associated with the selection of the viscosity, so that the sedimentation of particles in the film casting process does not occur too quickly.

The viscosity of the casting solution must be selected according to the height of the casting gap, and the rule is that the thicker the layer of the applied solution, the higher the required viscosity. A 15% HPMC type 606 solution and a 15% solution of a mixture of AP05, AP40, and PVP in a ratio of 2.5:5.0:7.5 were selected for the incorporation of the micronized LO particles. In our preliminary studies, we found that the films with high PVP content were too brittle and a higher ratio of AP led to formation of the films that were too elastic but also too fragile. Combination of PVP and AP in the proposed ratio resulted in sufficient elasticity of the films provided by AP and reduced disintegration time due to presence of PVP. It was also important that at the tested concentration of the polymers in the casting solution, i.e., 15% *w/w*, the compositions containing AP05:AP40:PVP in the ratios of 5:2.5:7.5 or 4:3.5:7.5 were not enough viscous (viscosity 0.26 and 0.35, respectively). The viscosities of the HPMC and AP/PVP solutions selected for further studies were 1.9 and 0.76 Pa s, respectively. Such masses did not allow, however, for the formation of a 1000 µm gap of placebo films, characterized by a uniform thickness and smooth surface, but the use of a smaller gap (800 µm and less) resulted in the proper formation of the films. More viscous solutions were disadvantageous due to the problem of de-aeration and a less even distribution of the mass on a plate. It was found that the viscosity of the proposed matrix compositions also allowed for the preparation of a homogeneous LO suspension, without rapid sedimentation. This is in agreement with previous publications. Woertz et al. [17] recommended for ODFs concentrations of various cellulose derivatives (HPMC 606 and HPC JXF) in a range from 7.5% to 18%, which corresponded to the viscosities between 0.5 and 14 Pa s.

LO was introduced into the viscous solutions containing 15% *w/w* of polymer and 3% *w/w* of PEG in the form of micronized powder, in which 96% of the particles had a size below 10 μm. The final LO content in the ODF was 10%–40% *w/w*. The incorporation of solid micronized LO particles caused a concentration-dependent increase in the viscosity (Figure 1). The relative increase in the viscosity (in comparison to the placebo solution) was similar for both types of the matrices and depended only on the concentration of dispersed solids.

A relatively high mass viscosity requires a proper LO dispersion method. Three different laboratory methods, using an Ultra-Turrax homogenizer, ultrasonic immersion probe, and planetary mixer, were compared (Table 1). The use of an Ultra-Turrax stirrer (15 min, 12 000 rpm) did not allow the preparation of homogeneous LO dispersions, and in the microscope image (Figure 2), aggregated particles up to 660 µm in size were visible. Moreover, the films obtained were characterized by an uneven surface. Ultrasonic homogenization eliminated the LO aggregates, but during the process, a premature gelation on the probe and non-uniform consistency in the mass were observed. Due to these changes in the mass structure, no further film was formed.

Only the use of the Thinky planetary mixer in deaeration mode (2 200 rpm, 15 min) ensured a homogeneous dispersion and proper quality of the prepared suspension (Figure 2). This homogenization method largely removed air bubbles, but additional de-aeration using a vacuum chamber or storage in a refrigerator (at least for 2 h) was required for the complete removal of air vesicles.

The ODF formed by casting onto a glass support plate, which had a smooth and shiny lower surface while the upper surface was rougher. Differences in the structure of both surfaces were demonstrated using AFM (Figure 3) and digital (Figure 4) microscopes. It is visible that in the presence of LO particles, the bottom surface of the film was not ideally smooth, however the roughness was very small—up to 1.4 µm. With a uniform distribution of LO, the upper surface was also relatively smooth: A maximum roughness of 5 µm was determined. Preliminary findings (studies performed with three probants) showed that the feeling of roughness in the case of the incorporation of micronized particles (even up to 40% LO) was low and should ensure an adequate acceptance of application by patients.

### 3.2. Mechanical Properties of ODFs

A critical quality attribute of ODFs is the mechanical strength, enabling both a correct production and packaging process, as well as ensuring the ease of application and removal of the film from the packaging during use [37]. Films must be flexible enough not to crumble or crack during manufacturing (on roller feeders, when forming, and moving film sheets) and when the patient removes them from the packaging. Excessive flexibility, however, is disadvantageous due to the possibility of a loss of shape, which may even lead to tearing, when removed from the package.

ODFs prepared with AP/PVP and HPMC exhibited different viscoelastic behavior in a tensile test, and for AP/PVP films, a much higher elongation at break was observed, which demonstrates that this film is more elastic (Figure 5). In both types of ODFs, a yield point (limit of elasticity) was observable. In contrast to HPMC, the AP/PVP films did not break easily, but instead developed a neck, and during the following stage (plastic region) of the test, a slope of the stress–strain curve was still observed, while in the plastic region, HPMC films elongated under constant stress. Figure 5 also clearly demonstrates that the LO particles incorporated in ODFs decreased their tensile strain in a concentration-dependent manner. This behavior is explained by the fact that the solid particles incorporated in the polymer film disturb the matrix continuity, which led to a decrease in the tensile strength [39,40]. However, only in HPMC films is the effect of LO on the yield observed.

All examined films at low stress values showed a linear stress–strain relationship, which allowed for the determination of Young’s modulus (E). High values of E characterize brittle films. Thus, according to this parameter, the ODF with HPMC, not only the placebo, but also with LO, have a fairly rigid structure that does not deform, e.g., when removed from plates, while maintaining a certain flexibility, depending on the LO concentration, thus allowing for convenient application. Placebo ODFs containing up to 15% LO, obtained from AP/PVP, were characterized by a much greater flexibility than those prepared with HPMC, as evidenced by the high value of %E. However, for these films, there was a large reduction in the elasticity at a higher LO concentration (Table 2, Figure 5). Regardless of the concentration of solid particles introduced (up to 30%), AP/PVP films did not crumble and were resistant to folding. The higher hardness and lower flexibility of ODF with HPMC was also demonstrated by the lower folding strength (Table 2). As the proportion of solids in this matrix increased, however, the folding strength decreased significantly.

After the addition of LO, the ODF HPMC flexibility decreased significantly, but the %E parameter was similar for films with a LO content of 10%–30%. The films made of HPMC, incorporating solid particles, showed a greater stability in terms of mechanical properties. It is true that the introduction of LO even at a concentration of 10% reduced the values of Young’s modulus, TR, and %E, but the decrease in these values was relatively small (10%–30%). A further increase in the concentration of solid particles, from 10% to 30%, did not change the mechanical parameters. Unacceptable changes were observed if the LO content was 40% *w/w*. Therefore, due to the mechanical properties of the films, the upper limit of the concentration of the incorporated micronized particles was 30% *w/w*.

Values of %E that are too high (340%–260%), which are observed for AP/PVP films at a low LO load (up to 15% *w/w*), make it difficult to package or divide the film into single-dose units, due to deformation. Although there are no limits defined for the viscoelastic parameters that correlate with a good performance of ODF during manipulation, our observations and some data reported by other authors may suggest that the values of %E below 100% are usually applicable [41]. For products on the market, values %E below 40% were measured [42]. While a greater film flexibility may be beneficial in terms of organoleptic sensations associated with application in the oral cavity, films that are too elastic, like those with an LO content of less than 20%, may be ductile, e.g., during removal from the packaging. With respect to pilot- and production-scale manufacturing, elasticity might lead to unintended stretching of the films, e.g., during the transfer over conveyor belts, which may result in waving of the thin film due to rebound effects and yield irregular drug content. Brittle films, on the other hand, may provoke ruptures during production and cutting [15].

The mechanical properties of ODFs may depend on the water content. The water content, tested by the Karl–Fisher method for the HPMC and AP/PVP placebo films, was 3.0% and 5.8%, respectively, and at a 30% LO content, it was 2.2% and 4.1%, respectively. The observed differences were small, which allows the conclusion that the different mechanical properties of the prepared ODFs result from the use of polymers of different structures and properties. It is worthy to note that a low water content is beneficial for chemical and microbiological product stability.

### 3.3. Disintegration Time

Two types of film were initially tested: One made with a 300 µm gap, and another made with an 800 µm gap. The placebo ODF thickness, obtained with the 800 µm gap, was about 100 µm (Table 2), whereas 30–35 µm thick films were obtained using the 300 µm gap. The results of the dependence of the mass of the obtained films on the polymer used and the concentration of the incorporated solids are presented in Figure 6. Films made from AP/PVP showed a greater mass than those prepared with HPMC, which resulted from a certain difference in the thickness. In both cases, the thickness and weight of the films also increased with the increase of the LO content (Table 2, Figure 6).

The films made using a 300 µm gap, without API and containing 10% *w/w* of LO, disintegrated very fast (in about 10 s), regardless of the type of polymer used. The thickness of these films was only 30-35 µm, and because of the insufficient API dose that might be potentially incorporated, the development of such thin films with LO was not continued. Figure 7 shows the disintegration time of ODFs with a thickness of 100-150 μm, obtained using an 800 μm gap. Increasing the ODF thickness resulted in an extension of the in vitro disintegration time, depending on the matrix polymer used, to approximately 110 s and 35 s for HPMC and AP/PVP, respectively.

The incorporation of micronized solids did not extend the disintegration time of ODFs prepared from HPMC, and in the case of a LO content of 10%–20%, an even shorter disintegration time was noted (*p* = 0.004). These small differences may not be significant for patients in vivo. In the case of ODFs prepared using AP/PVP, the introduction of solid particles in a concentration of up to 30% extended the disintegration time by up to 2 times, but it was still below 60 s and shorter than that in the case of the ODF with HPMC. The observed clear differences between two types of the matrices are difficult to explain. One of the possible reasons may be associated with the balance between hydrophobic nature of LO particle and hydrophilicity of the matrix polymer and mechanism of its dissolution. The solid particles introduced to a very fast disintegrating AP/PVP film may decrease its wettability. This effect, however, may not be visible when disintegration requires more time, like in HPMC film. 

Although a simple disintegration test may not correlate very well with the in vivo results, it is worthy to notice that the observed differences prove a significant discriminating power of such test if the aim is to compare formulations and to conclude on the effect of different technological factors.

### 3.4. Uniformity of Drug Content

The degree of identity in the amount of active substance in single-dose ODFs prepared using the 800 µm gap was evaluated on the basis of a pharmacopoeial method for the uniformity of dosage units. The obtained results, average values, and calculated AV values are presented in Figure 8. LO was incorporated at a dose of 1 mg/cm^2^ (10% *w/w*) to 6 mg/cm^2^ (40% *w/w*).

A greater homogeneity of the LO solids dispersion and dosing accuracy were obtained for the formulations made using HPMC: The average LO content in the examined films was within the required limits of 85%–115%, and the SD value did not exceed 4.6%. Homogeneous films with an AV acceptance value of <15 were obtained, regardless of the LO concentration introduced (10%–40%). The films from AP/PVP were characterized by a higher average LO content, from 102.7% to 112.5%, and the SD value was up to 7.5%. The required AV values of <15 were met only for LO at a concentration of 10% *w/w*.

## 4. Conclusions

It was found that 15% HPMC solutions or mixtures of AP and PVP have a viscosity suitable for obtaining a homogeneous LO suspension, from which ODFs can be obtained using a gap of 800 µm or less. It was shown that a high-speed planetary Thinky mixer most effectively homogenized LO suspensions in a viscous polymer solution, with little aeration. The roughness of film containing up to 40% micronized loratadine was low. The mechanical properties of the obtained films depended on both the LO concentration and the type of matrix polymer. The incorporation of micronized particles leads to a reduction in the mechanical strength of films to a greater extent in the case of HPMC than in the case of AP/PVP. Changes in the mechanical properties, resulting from the presence of small particles, with a concentration of up to 30% *w/w* of the final ODF mass, do not adversely affect the applicability of the resulting films. It is possible to obtain HPMC ODFs characterized by an adequate uniformity of the content of LO, incorporated in a concentration from 10% to 40%, while a lower uniformity of content was obtained in the case of the ODF with AP/PVP. In effect, despite the shorter disintegration time of the ODF with AP/PVP, the matrix composed of HPMC was selected as the more suitable for incorporation with LO in a concentration of 30% *w/w* to obtain 5 mg of API in an ODF of 1.5 cm^2^ in size, with an in vitro disintegration time below 100 s.

## Figures and Tables

**Figure 1 pharmaceutics-12-00250-f001:**
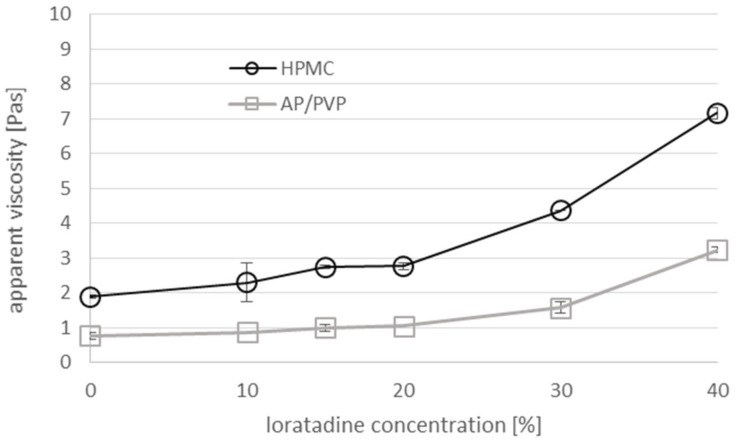
The effect of the concentration of suspended Loratadine (LO) (the final concentration in the ODF of 10%–40% *w/w* corresponds to 2%–12% in the casting mass) on the apparent viscosity of the hypromellose (HPMC) and polyvinyl alcohol (AP)/polyvinyl alcohol povidone (PVP) casting mass (rotational rheometer, shear rate: 10 s^−1^, *n* = 3, mean ± SD).

**Figure 2 pharmaceutics-12-00250-f002:**
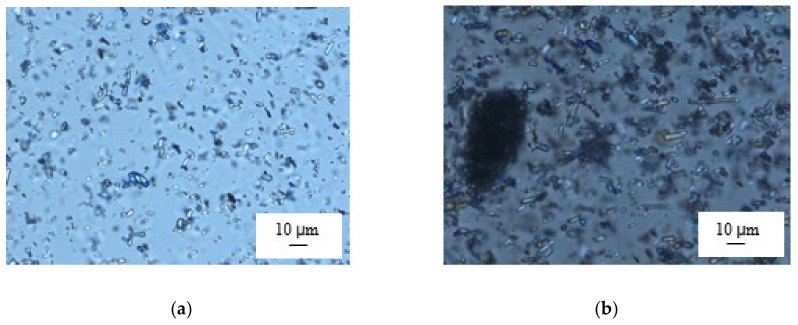
Microscopic image of the LO suspension (30% *w/w*) in the HPMC casting mass homogenized in a planetary mixer, Thinky (**a**), or in Ultra-Turrax (**b**); the scale bar is 10 µm.

**Figure 3 pharmaceutics-12-00250-f003:**
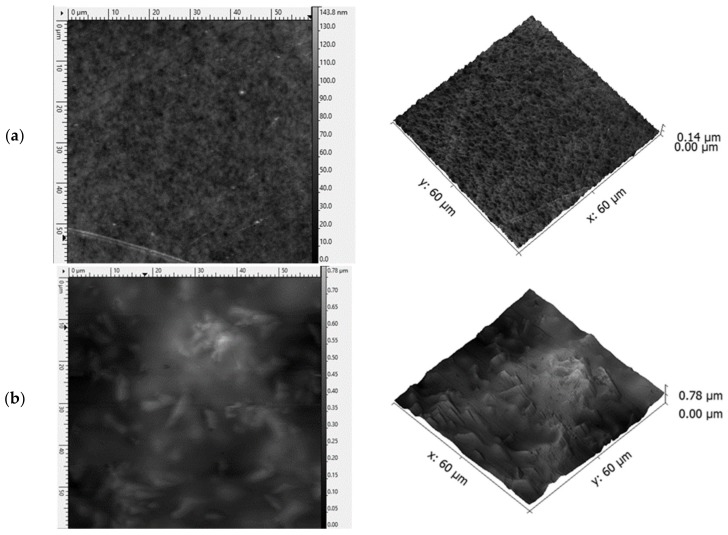
Atomic force microscopy (AFM) images (2D) and topography (3D) of HPMC ODF: (**a**) Upper surface of a drug-free film, (**b**) lower surface of an LO film, (**c**) upper surface of an LO film (30% *w/w* of LO in casting mass).

**Figure 4 pharmaceutics-12-00250-f004:**
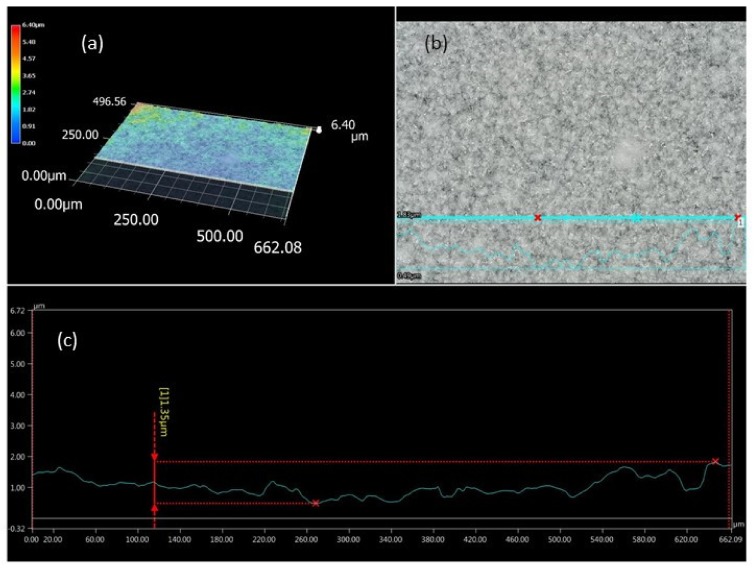
Digital microscope images (**a** and **b**) and 3D surface topography (**c**) of the upper surface of the HPMC ODF (LO 30% *w/w*).

**Figure 5 pharmaceutics-12-00250-f005:**
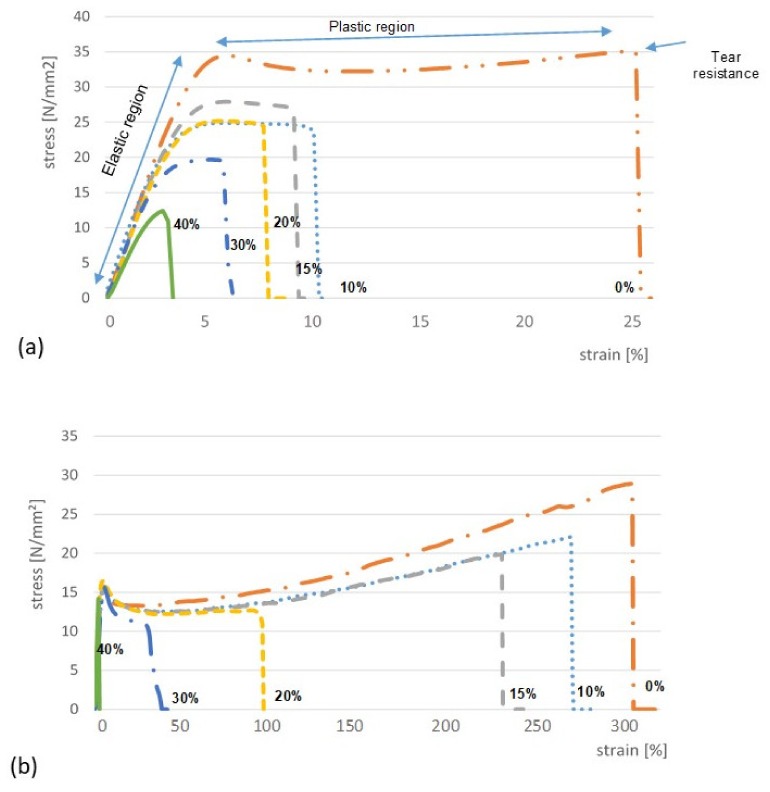
The influence of different loadings of solid particles (LO 0%–40%) on stress–strain diagrams in relation to the polymer type in ODFs: (**a**) HPMC 606 and (**b**) AP/PVP.

**Figure 6 pharmaceutics-12-00250-f006:**
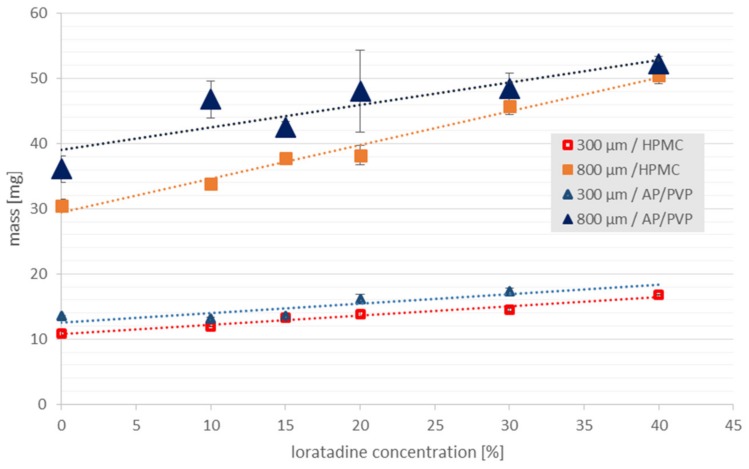
Correlation between solid particles loading (LO 10%–40% *w/w*) and the mass of the ODFs (3 cm^2^) prepared with the HPMC or AP/PVP matrix (casting heights of 300 µm and 800 µm).

**Figure 7 pharmaceutics-12-00250-f007:**
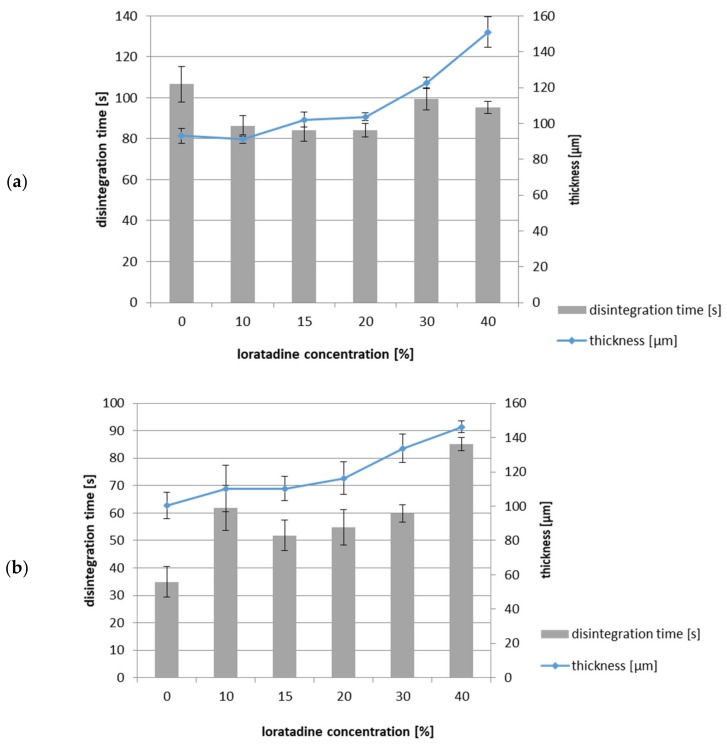
Influence of solid particles loading (LO 0–40% *w/w*) on the disintegration time (*n* = 3, mean ± SD) and film thickness (*n* = 10, mean ± SD) of ODFs prepared with a casting height of 800 µm: (**a**) HPMC, (**b**) AP/PVP matrix.

**Figure 8 pharmaceutics-12-00250-f008:**
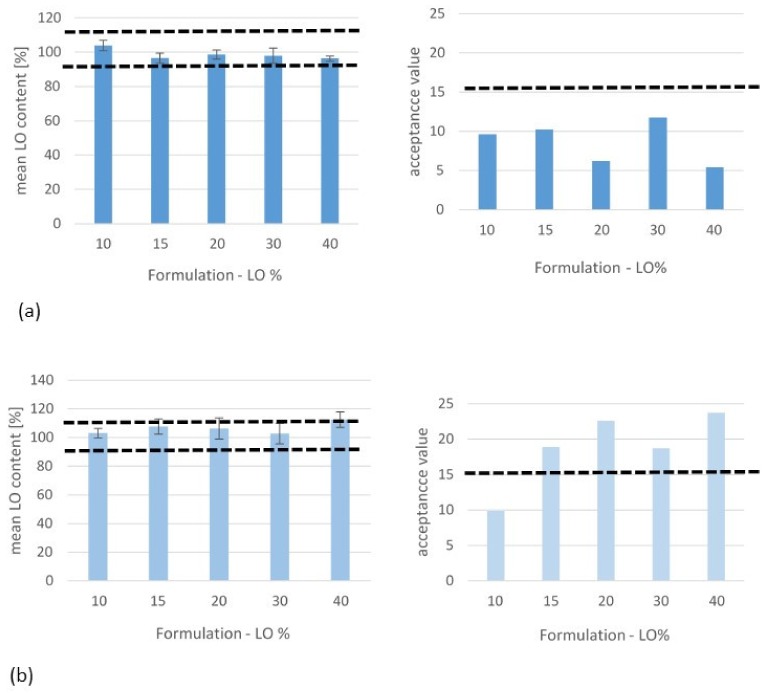
Uniformity of the LO content in ODFs: % of the theoretical dose in a single ODF (*n* = 10, mean ± SD) and acceptance values (AV). ODFs were prepared with HPMC (**a**) or AP/PVP (**b**), with different LO loadings (10%–40% *w/w*) and a casting height of 800 µm.

**Table 1 pharmaceutics-12-00250-t001:** Comparison of the homogenization methods (TM—planetary mixer Thinky; UT—Ultra Turrax; US—probe type ultrasonic disintegrator). The effects on homogeneity of the casting mass and the prepared orodispersible films (ODFs) with LO (30% *w/w*).

Method	Casting Dispersion		Film Morphology	
	HPMC	AP/PVP	HPMC	AP/PVP
**TM**	homogenous, degassed	homogenous, degassed	homogenous, peelable, elastic	homogenous, peelable, very elastic
**UT**	non-homogenous (LO aggregates up to 660 µm), highly aerated	non-homogenous (LO aggregates up to 220 µm), aerated	non-homogenous, rough upper surface, aerated, peelable, elastic	non-homogenous, rough upper surface, aerated, peelable, very elastic
**US**	local non-homogenous gelation, degassed, LO homogenously dispersed	local non-homogenous gelation, degassed, LO homogenously dispersed	unformable	unformable

**Table 2 pharmaceutics-12-00250-t002:** Influence of solid particles loading (LO 0%–40%) on the mechanical properties of ODFs in relation to the polymer type: HPMC 606 and AP/PVP (*n* ≥ 6, mean ± SD).

Polymer	LO (%)	Thickness * (µm)	Young’s Modulus E (N/mm^2^)	Tear Resistance TR (N)	Elongation at Break %E (%)	Folding Endurance FE
**HPMC**	0	93.0 ± 4.1	771.3 ± 26.5	33.4 ± 1.5	26.2 ± 0.85	> 200
	10	91.2 ± 2.5	526.3 ± 30.3	22.3 ± 0.3	6.3 ± 0.3	> 200
	15	102.1 ± 4.2	557.0 ± 16.1	27.2 ± 1.1	6.0 ± 0.4	39
	20	103.7 ± 2.2	522.4 ± 33.4	26.2 ± 1.2	5.7 ± 0.3	12
	30	122.7 ± 3.0	396.3 ± 25.1	23.1 ± 0.5	5.6 ± 0.4	6
	40	150.9 ± 8.8	449.6 ± 55.6	17.6 ± 0.3	3.1 ± 0.1	1
**AP/PVP**	0	100.40 ± 7.8	300.0 ± 26.1	27.6 ± 2.6	338.5 ± 10.3	> 200
	10	110.30 ± 13.5	325.2 ± 18.9	28.2 ± 2.1	284.9 ± 20.5	> 200
	15	110.30 ± 6.9	339.6 ± 25.4	20.3 ± 0.8	257.5 ± 6.9	> 200
	20	116.30 ± 9.6	321.3 ± 11.7	20.1 ± 0.6	5.3 ± 0.1	> 200
	30	133.70 ± 8.3	343.1 ± 57.5	21.8 ± 1.4	5.0 ± 0.5	> 200
	40	146.30 ± 3.3	946.0 ± 77.6	21.5 ± 1.9	± 0.1	1

* the films casted with a gap of 800 µm.

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
