# Peer review of "Technology of Orodispersible Polymer Films with Micronized Loratadine—Influence of Different Drug Loadings on Film Properties"

_pharmaceutics, 2020, doi:10.3390/pharmaceutics12030250_

Round 1

Reviewer 1 Report

The manuscript „Technology of orodispersible polymer films with micronized loratadine – influence of different drug loadings on film properties” handed in by Centkowska et al. has a clear structure and characterizes the manufactured ODFs with the commonly used methods. Please find my comments below:

  • Line 22: What does it mean that the films are “too elastic”? Please comment.
  • Line 208: Where does the ration of 2.5:5.0:7.5 come from? Literature or former studies? Please refer to former studies or give more information why this particular ration was chosen.
  • Line 231: “The use of the Ultra-Turrax stirrer did not allow for the preparation”… “and aggregated particles up to”…
  • Line 231: How were the sizes of the aggregates measured, which method was used? Please comment on this.
  • Line 242: What does it mean that the suspensions were stored at “low” temperatures? Can you please give the exact temperature and a reason why they were stored at these temperatures?
  • Figure 2: The scale in the pictures is too small, please enlarge them.
  • Line 249: This sentence sounds like not all ODFs were casted on a glass plate. Could you please comment on this and/or rephrase the sentence?
  • Line 253: When casting ODFs on a plate it is expected to get a very plain bottom surface, because of the polymer solution which is used as matrix for the film. Could you please explain why you detected a clear roughness for the lower surface of the LO films, also indicated in the image of Figure 3 (b)?
  • Line 188 and 268: The headings are similar, please rephrase 3.2
  • Line 275: “ODF films”, ODF = orodispersible film
  • Line 284: The reference [41] is missing in the literature list at the very end of the manuscript. Please check and add the reference.
  • Line 292 – 331: The information in Figure 5 and Table 2 is quite similar. Please combine the discussions for the mechanical properties of the films to avoid repetitions.
  • Line 313: You refer to Figure 4 in your manuscript, shouldn’t that be Figure 5?
  • Line 313: What does “too high” mean? Can refer to a reference in literature? Up to which Young’s modulus are the films of high quality? Please comment on this.
  • Line 352-357: Clear differences can be detected between HPMC and AP/PVP films when particles are incorporated in the film matrix. What is the reason for this? Please discuss and compare to literature.
  • Line 369-375: How was the sampling of these films? Did you collect the samples from all over the film in a fixed pattern?

Reviewer 2 Report

The article is interesting, well written and fits the aim and scope of the journal. The experimental campaign is well designed. The analysis of different dispersion methods is well addressed and interesting in the results.

I have two concerns.
One is related to the analysis of disintegration time or, better to say, to the procedure adopted to evaluate the disintegration time that is a little bit rough and, from my point of view, does not allow for a reliable estimate of the disintegration time and how it is influenced by Lotadine loading. From data reported in Fig. 7 I can conclude that the disintegration time is very small, an order of 100 s for both formulations and almost independent of Lotadine concentration. I would like the author to add a comment on this.

The second concern is related to the analysis of rheological measurements. It would be nice if the authors include some flow curves and data for G' and G'' to better understand the influence of Lotadine concentration on rheological properties of the casting solution.

A final small concern is related to the analysis of mechanical properties. Authors use a very large film 10x20 cm^2, if I am not wrong. Why such a big film? How they can guarantee a uniform drug distribution is such a big film and how this can influence the resulting mechanical properties? Again, I would like the authors to comment on this.

I'd like the authors to add some information on the swelling properties of these OTFs. If the disintegration time is that small, I understand that it is very difficult to have a reliable estimate of film swelling. However, HPMC can swell a lot and if the swelling is fast (as I expect), in the first 10-15 seconds the patient can experience a very bad sensation before the disintegration step takes place.

On line 307, the authors wrote that "the films casted with a gap of 800 m" ... is this a typo? I guess so.
